# Spatial Distribution Pattern of the Mesozooplankton Community in Ross Sea Region Marine Protected Area (RSR MPA) during Summer

**Sung Hoon Kim [1], Wuju Son [1], Jeong-Hoon Kim [2] and Hyoung Sul La [1,3,*]**

1    Division of Ocean & Atmosphere Sciences, Korea Polar Research Institute, Incheon 21990, Republic of Korea;
     ksungh13@kopri.re.kr (S.H.K.); swj5753@kopri.re.kr (W.S.)
2    Division of Life Sciences, Korea Polar Research Institute, Incheon 21990, Republic of Korea;
     jhkim94@kopri.re.kr
3    Department of Polar Science, University of Science and Technology, Daejeon 34113, Republic of Korea
*    Correspondence: hsla@kopri.re.kr

**Abstract:** The Ross Sea region Marine Protected Area (RSR MPA) is one of the most productive regions in the Southern Ocean. Mesozooplankton intermediates the primary product to the higher predators, such as penguins and seals, in this ecosystem. In this study, the mesozooplankton community structure and spatial pattern in the RSR MPA in January were investigated by using 505 μm-mesh-size bongo net samples. As a result, 37 mesozooplankton taxa with a total mean abundance of 35.26 ind./m$^3$, ranging from 2.94 to 139.17 ind./m$^3$, were confirmed. Of the 37 taxa, 7 occupied almost 84% of the total abundance, with copepods being the main dominant taxa. As shown by our hierarchical analysis, the mesozooplankton community was divided into four groups, each associated with a specific geographical distribution. Group A was composed of stations around Terra Nova Bay and showed relatively low abundance. Group B included stations around the continental slope region. Group D was composed of the Ross Sea continental shelf stations, while group C consisted of stations geographically located between those of groups B and D. These four groups were influenced by various environmental factors, such as water temperature, salinity, and nutrients. In summary, the mesozooplankton community can be separated according to geographical pattern. This pattern is related to several environmental factors.

**Keywords:** Ross Sea region Marine Protected Area (RSR MPA); zooplankton; January; Southern Ocean; community structure





## 1. Introduction

The Ross Sea comprises one of the largest continental embayments, with a depth range of 200 to 3000 m in the Southern Ocean [1]. Within the Ross Sea continental shelf, there are three troughs—the Drygalski Trough, Joides Trough, and Glomar Challenger Trough—that cross the continental shelf in a north–south direction [2]. Several water masses are transported through these three troughs. In other words, relatively warm Circumpolar Deep Water (CDW) flows into the Ross Sea continental shelf through these troughs [3,4]. Relatively cold and salty Shelf Water (SW), derived from Terra Nova Bay and the Ross Sea ice shelf, flows out of the continental shelf [4,5]. Consequently, CDW and SW mix and make modified Circumpolar Deep Water (MCDW) and modified Shelf Water in the Ross Sea continental shelf [5]. The salty SW, formed in the Terra Nova Bay and Ross Sea ice shelf, is one of the characteristics of the Ross Sea [6]. Furthermore, there are two coastal polynyas, the Terra Nova Bay polynya and Ross Sea polynya, in the Ross Sea continental shelf region [5]. These two coastal polynyas expand during summer and contribute to higher primary productivity [5,7,8]. This high primary productivity attracts various organisms that consume primary producers, from zooplankton to penguins and seals [9]. Among

them, mesozooplankton such as copepods play a crucial role in the energy transfer from primary producers to higher consumers [7,10,11]. However, no surveys have covered the entire Ross Sea region, including the Ross Sea continental shelf and Terra Nova Bay. The previous surveys of mesozooplankton in the Ross Sea were primarily conducted around McMurdo Sound or Terra Nova Bay, which are located near the McMurdo stations and Mario Zucchelli station [12–16], and the western Ross Sea [17–19]. Furthermore, scientific surveys and monitoring programs in this region have been required since the Ross Sea was designated as a Marine Protected Area (MPA) in December 2017 [20,21]. According to recent studies, the sea ice concentration has increased in the last year, contrary to the recent climate trend [22,23]. However, extensive ice cover and remoteness present significant logistic challenges that hinder detailed scientific research in this area [10,24].

Therefore, we surveyed the Ross Sea region Marine Protected Area (RSR MPA), covering the Ross Sea continental shelf and slope region, as well as Terra Nova Bay, during summer, mainly in January. This study aimed to describe the structure and spatial distribution pattern of the mesozooplankton community within the RSR MPA. Furthermore, the mesozooplankton community was assessed alongside environmental factors such as temperature, salinity, and nutrients to reveal the relationships between them.

## 2. Materials and Methods

This study was carried out through a survey in the RSR MPA on the Korean icebreaking research vessel (IBRV) *Araon* from January 17 to 1 February 2023. Mesozooplankton samples were collected from 22 stations ranging between 296 and 2694 m in depth (Figure 1). Environmental factors were also measured at the same sampling stations.

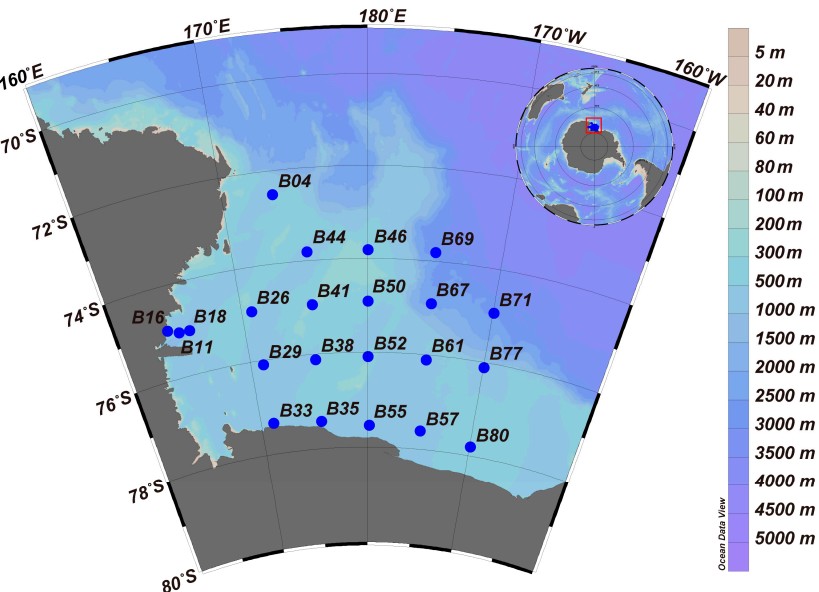

**Figure 1.** Map showing the survey region (red frame) and sampling stations (blue circles) within the RSR MPA from 17 January to 1 February 2023.

### 2.1. Data Sampling and Sample Processing

Mesozooplankton were vertically sampled from an approximate depth of 200 m using a bongo net with a mesh size of 505 µm (mouth diameter: 0.6 m) equipped with a flow meter. The collected samples were subdivided into two subsamples using a Folsom plankton splitter. One subsample was immediately fixed in 4% seawater formalin on board, while the other subsample was stored in a deep freezer at −70 °C for further analysis, such as DNA and fatty acid analysis. The fixed samples were transported to the laboratory and identified based on morphological characteristics under a dissecting microscope (Zeiss STEMI SV8). If necessary, specimens were dissected and observed under a compound microscope

(Olympus BX51). Samples were identified at the species level where possible, although several taxa, such as Appendicularia, Ostracoda, and polychaete, were not identified due to species identification difficulties. Additionally, copepodite developmental stages were not distinguished in this study. Abundance was converted to individual numbers per cubic meter (ind./m$^3$), estimated by multiplying the mouth area of the bongo net by the depth reached (200 m).

For the environmental analyses, vertical profiles of temperature (Temp, °C), salinity (Sal, PSU), and dissolved oxygen (DO, mg/L) were obtained from the surface-to-bottom depth of each station using a conductivity–temperature–depth (CTD) probe (Sea-Bird, SBE 911 plus) with an oxygen sensor attached at every sampling station. To measure chlorophyll-*a* (Chl-*a*, μg/L) and the concentrations of nutrients (μM) such as $PO_4$, $NO_3 + NO_2$, and $SiO_2$, water samples were sampled using a 10-L PVC Niskin bottle rosette affixed to the CTD probe's frame. The sampled water was filtered through a cascade connection of 20 μm nylon mesh, a nuclepore filter (Whatman International) with a pore size of 3 μm, and then a Whatman GF/F filter to determine size-fractionated Chl-*a*. Consequently, the size-fractionated samples were categorized as Chl-*a*$_{Micro}$ (>20 μm), Chl-*a*$_{Nano}$ (3 to 20 μm), and Chl-*a*$_{Pico}$ (0.7 to 5 μm) [25]. Another sample of water collected for Chl-*a* analysis was filtered through 47 mm GF/F Whatman filters. Chlorophyll-*a* concentrations were measured by using a fluorometer (model: Trilogy, Turner Design, USA) after extraction using 90% acetone in the dark for 24 h [22]. For our nutrient analyses, the water samples were promptly stored in a refrigerator at 4 °C and analyzed within 2 days. Each nutrient concentration was determined using standard colorimetric methods adapted for use on a 4-channel continuous Auto-Analyzer (QuAAtrp, Seal Analytical).

For subsequent analyses, Temp, Sal, and DO were modified to the surface and average values (e.g., Sal$_{surface}$ and Sal$_{average}$). The average values were estimated from the surface to 200 m. However, the chlorophyll-*a* and nutrient ($PO_4$, $NO_3 + NO_2$, and $SiO_2$) values were applied only as average values estimated from the surface to 100 m for chlorophyll-*a* and from the surface to 200 m for nutrients.

### 2.2. Data Processing

All statistical analyses were conducted using the PRIMER v7 package [26,27] and PERMANOVA + for PRIMER [28]. In some cases, the Ocean Data View (ODV) software package was utilized to visualize the spatial patterns of mesozooplankton and environmental parameters. For the environmental analyses, draftsman plots were constructed to apply an appropriate transformation to the environmental parameters [29]. As a result, DO$_{average}$, Chl-*a*$_{Total}$, Chl-*a*$_{Micro}$, and Chl-*a*$_{Nano}$ were transformed using log (X + 1) before analysis. After the transformation, all environmental parameters were normalized using the PRIMER v7 package before analyses. For the biotic analyses, mesozooplankton abundance was square-root-transformed prior to subsequent analyses. A resemblance matrix was established using Bray–Curtis similarity to identify similarities between the sampling stations. Hierarchical cluster analysis (CLUSTER) was performed based on the Bray–Curtis similarities by using group-average linking. PERMANOVA tests were conducted to determine whether there were significant ($p < 0.05$) differences among the grouped mesozooplankton communities. Similarity percentage (SIMPER) analysis was performed to verify which species contributed to the group similarities and dissimilarities. Canonical analysis of principal coordinates (CAP) was performed to confirm the patterns of divided groups and the correlations between the mesozooplankton communities and environmental parameters. The biological–environmental (BIOENV) procedure was followed to confirm which environmental parameters best explain the differences in the mesozooplankton communities.

## 3. Results

### 3.1. Spatial Variation in the Environmental Parameters

The values of the environmental parameters at the sampling stations are summarized in Table 1 and depicted in Figures 2 and 3. Considering sea ice concentration, Terra Nova Bay was surrounded by sea ice, while the Ross Sea was not (Figure S1). When comparing environmental parameters, the highest $Temp_{surface}$ was 0.44 °C at station B16, while the lowest was −1.61 °C at station B26. Similarly, the highest $Temp_{average}$ was −1.03 °C at station B41, whereas the lowest was −1.82 °C at station B18. Overall, the average water temperature was lower than the surface water temperature. Therefore, the average salinity was higher than the surface salinity. Salinities ranged from 33.72 to 34.72 PSU. Station B69 showed the lowest salinity value compared to the average values, while station B18 indicated the highest salinity value. The average salinity was higher than the surface salinity. However, DO values were higher at the surface compared to the average values. Station B44 had the lowest DO value for both the surface and average values, while station B11 at the surface and B35 on average had the highest DO values, respectively. Regarding the nutrient concentrations, $PO_4$ and $NO_3 + NO_2$ showed similar trends, with high values being observed along the northeast stations and relatively low values being observed around the southern stations (Figure 2). However, the $SiO_2$ values showed a relatively even distribution pattern across all stations, although some stations indicated relatively high or low values. For the chlorophyll-*a* concentration, there was a relatively high concentration observed near the Ross Sea continental shelf stations (Figure 3). However, the ratio of the size-fractionated concentration differed according to the region. The micro-size chlorophyll-*a* ($Chl\text{-}a_{Micro}$) concentrations were relatively high around the Ross ice shelf stations, whereas the nano-size chlorophyll-*a* ($Chl\text{-}a_{Nano}$) concentrations were relatively high near the Tera Nova Bay stations. Pico-size chlorophyll-*a* ($Chl\text{-}a_{Pico}$) was consistently low at all stations. Overall, the Terra Nova Bay stations exhibited relatively low temperatures and high salinity. The Ross Sea continental shelf stations showed high temperatures and DO values. High concentrations of nutrients were found around the continental slope stations. The concentrations of chlorophyll were elevated around the Ross Sea continental shelf stations, although the composition of chlorophyll-a differed according to the station and region.

**Table 1.** Summary of the environmental parameters sampled during the survey in the RSR MPA. Min values mean lowest values among all sampling stations, while Max values mean highest values.

| | Min | Max | Av | SD |
|---|---|---|---|---|
| $Temp_{surface}$ (°C) | −1.61 | 0.44 | −0.59 | 0.46 |
| $Temp_{average}$ (°C) | −1.82 | −1.03 | −1.42 | 0.21 |
| $Sal_{surface}$ (PSU) | 33.72 | 34.3 | 34.08 | 0.16 |
| $Sal_{average}$ (PSU) | 34.08 | 34.72 | 34.40 | 0.18 |
| $DO_{surface}$ (mg/L) | 342.52 | 401.38 | 370.36 | 16.68 |
| $DO_{average}$ (mg/L) | 288.64 | 335.43 | 310.59 | 10.87 |
| $Chl\text{-}a_{Total}$ (µg/L) | 0.48 | 2.96 | 1.27 | 0.59 |
| $Chl\text{-}a_{Micro}$ (%) | 26.36 | 83.32 | 59.92 | 14.66 |
| $Chl\text{-}a_{Nano}$ (%) | 10.90 | 54.63 | 27.85 | 11.20 |
| $Chl\text{-}a_{Pico}$ (%) | 3.78 | 22.28 | 12.23 | 4.96 |
| $PO_4$ (µM) | 1.47 | 2.03 | 1.81 | 0.17 |
| $NO_3 + NO_2$ (µM) | 19.90 | 29.98 | 26.30 | 2.90 |
| $SiO_2$ (µM) | 61.43 | 78.65 | 72.82 | 3.86 |

Av: average; DO: dissolved oxygen; Max: maximum; Min: minimum; Temp: temperature; Sal: salinity; SD: standard deviation.

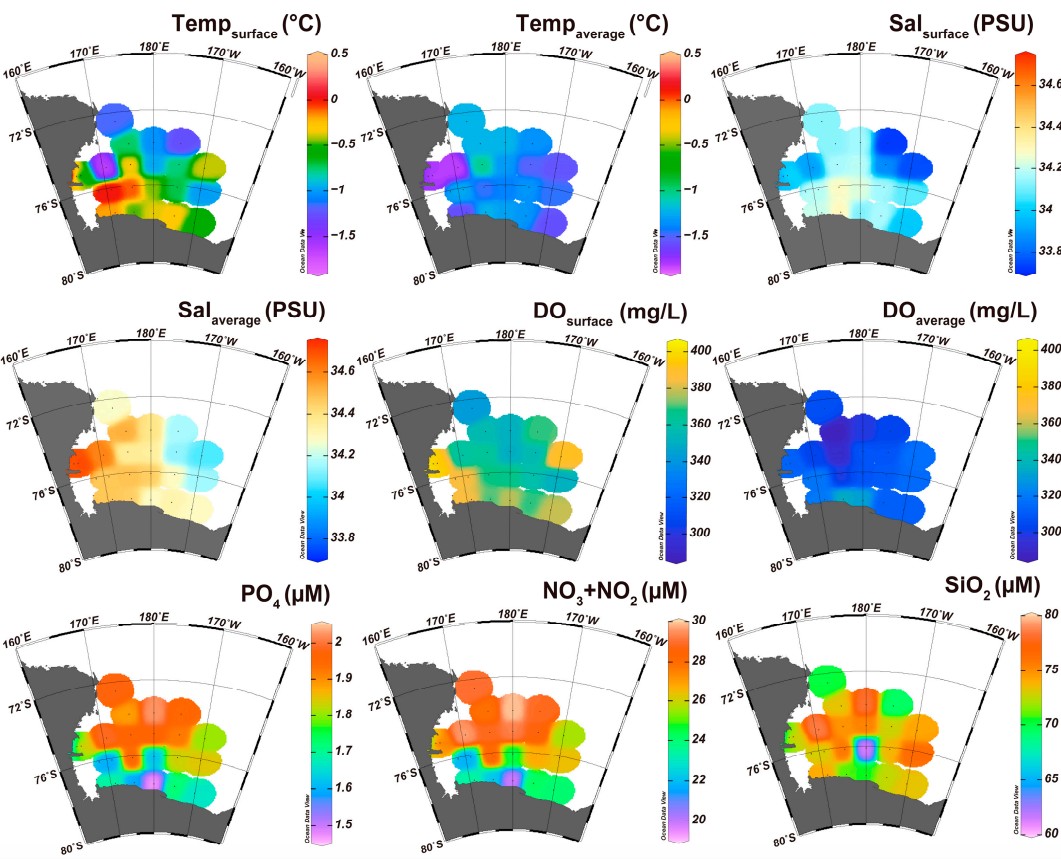

**Figure 2.** Spatial pattern of the environmental parameters in the RSR MPA.

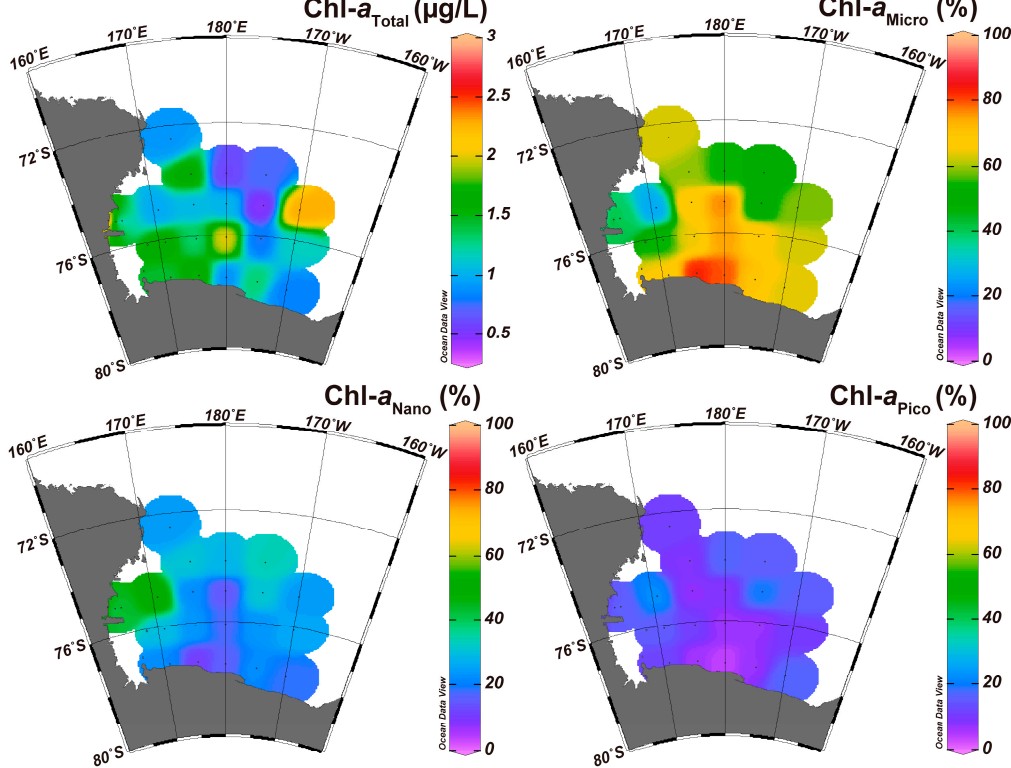

**Figure 3.** Spatial patterns of the chlorophyll concentrations in the RSR MPA. Size-fractionated chlorophyll concentration represents the composition (%) in the total chlorophyll-*a*.

### 3.2. Mesozooplankton Community Composition

The number of taxa and abundance are summarized in Tables S1 and S2 and Figure 4. In this survey, 37 mesozooplankton taxa were confirmed. The total mean abundance was 35.26 ind./m$^3$, ranging from 2.94 to 139.17 ind./m$^3$ among the stations. Station B18 showed the lowest number of taxa (9 taxa) and abundance (2.94 ind./m$^3$), while station B04 indicated the highest number of taxa (26 taxa) and abundance (139.17 ind./m$^3$). Cyclopoida such as *Oithona* spp. and *Triconia antarctica* showed high abundance in B04. The stations located around Terra Nova Bay, namely, B11, 18, and 26, showed a relatively low number of taxa and abundance. Of the 37 taxa, 7 showed an abundance above 1 ind./m$^3$. Among them, *Metridia gerlachei* was the most dominant taxon, occupying 36.2% of the total abundance. Subsequently, *Calanoides acutus* (13.9%), *Euphausia* calyptopis (9.2%), *Othona* spp. (7.9%), *T. antarctica* (6.5%), Cirriped nauplii (5.36%), and *Paraeuchaeta antarctica* (4.3%) were sub-dominated. Overall, the above seven dominant taxa comprised about 84% of the total mesozooplankton abundance. Among the dominant taxa, copepods occupied almost 70% of the total abundance. Considering the spatial distribution patterns of the dominant taxa (Figure 4), although *M. gerlachei* was present throughout the sampling region, there was a slightly higher concentration around the continental slope region. *C. acutus* showed relatively low abundance in B41, 44, 46, and 50. *Euphausia* calyptopis was abundant along the Ross Sea ice shelf. Cyclopoida and Cirriped nauplius were abundant in the northwest Ross Sea region. *P. antarctica* showed relatively low abundance around the east Ross Sea region.

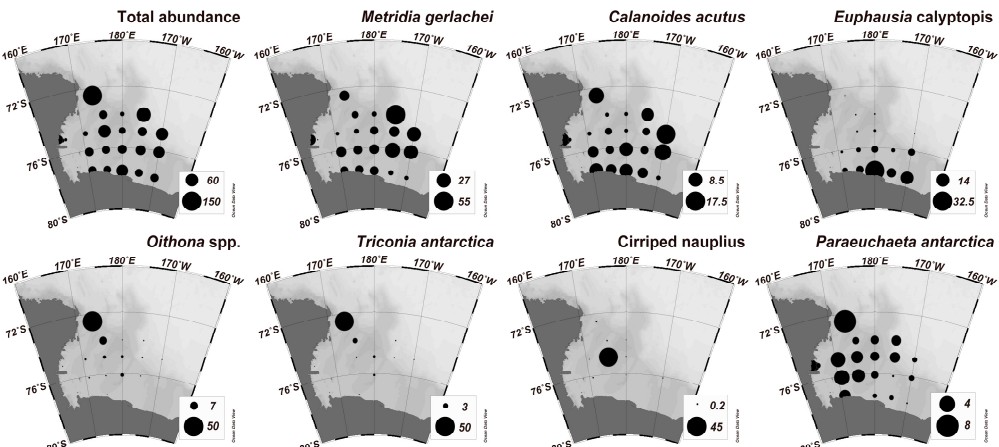

**Figure 4.** Comparison of the abundance (ind./m$^3$) values of the dominant taxa at each sampling station.

### 3.3. Spatial Pattern of the Mesozooplankton Community

In our hierarchical cluster analysis, the mesozooplankton community was divided into four groups (Figure 5). The first group (group A), separated at a 38.5% similarity level, consisted of three stations (B11, 18, and 26). The second group (group B), separated at a 48.9% similarity level, was composed of four stations (B04, 67, 69, and 71). The third group (group C), separated at a 49.17% similarity level, consisted of four stations (B41, 44, 46, and 50). The remaining stations were grouped into a fourth group (group D) at a 56.25% similarity level. Geographically, group A was located around Terra Nova Bay. Group B was positioned along the continental slope region. Group C included stations between the continental shelf and the slope region. Group D was composed of stations located on the continental shelf regions along the Ross Sea ice shelf. In the PERMANOVA test, the null hypothesis, stating that there were no significant differences in the mesozooplankton community between the divided groups, was rejected (pseudo-F = 5.9198, $p < 0.05$) (Table 2). Pairwise comparisons in the PERMANOVA test also showed significant differences between each pair of groups ($p < 0.05$) (Table 2).

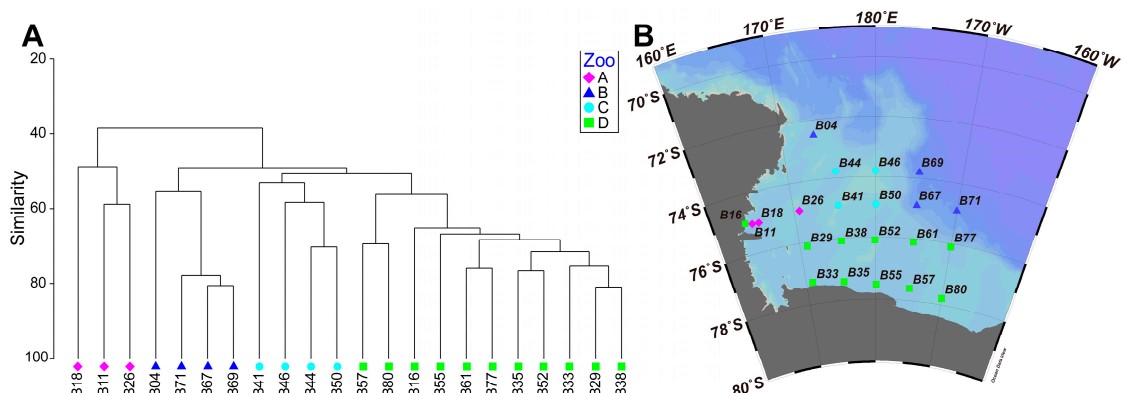

**Figure 5.** A dendrogram (**A**) based on the mesozooplankton abundance and a map (**B**) showing the separated groups based on the CLUSTER analysis.

**Table 2.** PERMANOVA results based on the mesozooplankton abundance data and comparisons between groups (pairwise tests). Significance levels below 0.05 were bolded.

| Source | df | SS | MS | Pseudo-F | Significance Level (*p*) |
|---|---|---|---|---|---|
| Groups | 3 | 13301 | 4433.6 | 5.9198 | **0.001** |
| Residual | 18 | 13481 | 748.94 | | |
| Total | 21 | 26782 | | | |

| Pairwise tests | df | t | Significance levels (*p*) | | |
|---|---|---|---|---|---|
| Group A and B | 5 | 2.7391 | **0.029** | | |
| Group A and C | 5 | 1.9677 | **0.025** | | |
| Group A and D | 12 | 2.6507 | **0.003** | | |
| Group B and C | 6 | 2.0928 | **0.033** | | |
| Group B and D | 13 | 2.6494 | **0.004** | | |
| Group C and D | 13 | 2.2354 | **0.001** | | |

In the SIMPER analysis, seven taxa contributed to the upper 70% of the group similarity between groups A and C, while five taxa contributed to the group similarity between groups B and D (Table S3). Calanoids, *C. acutus*, *M. gerlachei*, and *P. antarctica* made large contributions to all of the divided groups. The dissimilarity between the divided groups ranged from 49.23% to 69.26% (Table S4). Among them, the greatest dissimilarity was confirmed between groups A and B, whereas groups C and D showed relatively low dissimilarity. Thirteen taxa contributed to the upper 50% of the group dissimilarity. Among them, their abundance were low in group A. Cyclopoida showed relatively high abundance in group B. A high abundance of Cirriped nauplius was confirmed in group C. *Euphausia* calyptopis and *Limacina rangii* showed high abundance in group D.

When these divided groups were analyzed using CAP, the spatial distribution pattern was well divided into four groups (Figure 6). The first squared canonical correlation ($\delta^2 = 0.9007$) divided the mesozooplankton community into three groups: groups A, B + D, and C. The second canonical axis divided the mesozooplankton community into three groups: groups A + C, B, and D ($\delta^2 = 0.8884$). There was no classification error in the separation of the stations into four groups. When comparing the taxa showing a large contribution in the SIMPER analysis (Figure 6A), most taxa showed a positive correlation with groups B and C. However, *Euphausia* calyptopis and *L. rangii* indicated a highly positive correlation with group D. Fish larvae showed a highly positive correlation with group A. Regarding the environmental parameters (Figure 6B), $Temp_{surface}$, $DO_{average}$, $Chl\text{-}a_{Total}$, and $Chl\text{-}a_{Micro}$ showed a positive correlation with group D. Nutrients such as $PO_4$ and $NO_3 + NO_2$ showed a positive correlation with group C. Additionally, group A showed a positive correlation with $Sal_{average}$. Group B showed a positive correlation with $Temp_{average}$ while displaying a negative correlation with $Sal_{average}$. In the BIOENV

analysis, the mesozooplankton community was described by a combination of Temp$_{average}$, Sal$_{average}$, and PO$_4$ ($\rho = 0.512$, $p < 0.05$).

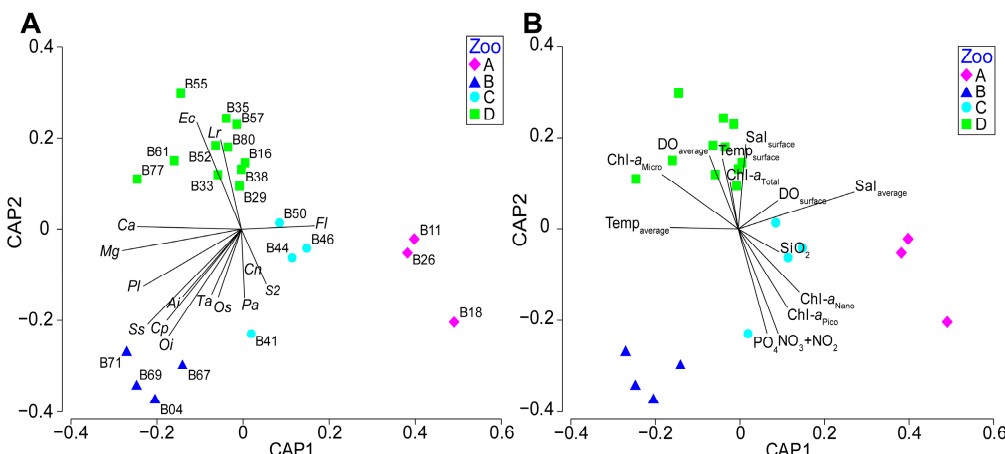

**Figure 6.** Canonical analysis of principal coordinates (CAP) plots according to the mesozooplankton abundance data showing correlations with the dominant species (**A**) and environmental parameters (**B**). Ai, Appendicularia indet.; Ca, *Calanoides acutus*; Cn, Cirriped nauplius; Cp, *Calanus propinquus*; Ec, *Euphausia* calyptopis; Fl, Fish larvae; Lr, *Limacina rangii*; Mg, *Metridia gerlachei*; Oi, Ostracoda indet.; Os, *Oithona* spp.; Pa, *Paraeuchaeta antarctica*; Pl, Polychaeta larvae; Ss, *Sagitta* sp.; S2, Siphonophore 2; Ta, *Triconia antarctica*.

## 4. Discussion

In the environmental analyses, water temperature and salinity showed similar distribution patterns except for the Terra Nova Bay region. These environmental parameters indicated an eastward gradient. In other words, the western stations of the Ross Sea region showed relatively high water temperature and salinity values, although the water temperature values of the Terra Nova Bay stations were lower than those of the Ross Sea continental shelf stations. Considering that Terra Nova Bay and the Ross Sea ice shelf are the main sources of Shelf Water (SW), which is generally cold and salty, these trends relied on the formation of the water masses [30]. Besides water temperature and salinity, nutrients such as PO$_4$ and NO$_3$ + NO$_2$ represented a geographical distribution pattern. These nutrients showed high concentrations around the continental slope region. Considering that Circumpolar Deep Water (CDW) has been found to flow along the Ross Sea continental slope and Modified Circumpolar Deep Water (MCDW) with nutrients has been found to flow into the northwest Ross Sea continental shelf region, these high concentrations of nutrients mean that valuable nutrients flow into the Ross Sea region along with MCDW through the northwest continental slope region [5,31]. Given that MCDW has a low concentration of DO, these results correspond well with the results of the present survey, which confirmed the presence of low DO near the Ross Sea continental slope region [10]. Regarding the concentration of chlorophyll-*a*, a relatively high concentration was confirmed around the Ross Sea continental shelf region, while the continental slope region showed a relatively low chlorophyll-*a* concentration. These results are in agreement with those of previous studies that have shown that primary production is high in the polynya regions [5,7,32].

Among 37 mesozooplankton taxa, 7 had abundance exceeding 1 ind./m$^3$ and constituted about 84% of the total zooplankton abundance. Among these seven taxa, five were copepods, occupying about 70% of the total mesozooplankton community. Consequently, the copepod taxon was the main constituent of the January Ross Sea mesozooplankton community. Our taxa composition list corresponds well with those of previous studies, indicating that copepods are the main constituent of the mesozooplankton community in the region [9,13,18,33]. Additionally, *M. gerlachei* was the most dominant species, followed by *C. acutus*. However, Cyclopoida such as *Oithona* spp. and *T. antarctica* showed

relatively low abundance. Considering that the mesh size of the sampling net affects the mesozooplankton community, the abundance of small zooplankton such as *Oithona* spp. and *T. antarctica* could have been underestimated in this study [34]. However, although the composition ratio differed according to the studies mainly affected by the mesh size of the plankton net used, the composition of mesozooplankton found in this study is similar to that of previous studies [13,15,20,35].

Considering the mesozooplankton community structure of the studied groups, group A, comprising the Terra Nova Bay stations, showed relatively low abundance except for station B16. Comparing the taxa composition, ice krill (*Euphausia crystallorophia*) and fish larvae showed relatively higher abundance than other groups. Overall, the low abundance and unusual taxa composition separated group A from the other groups. Considering that these stations showed much lower temperatures and higher nano-size chlorophyll-*a* concentrations than the other groups, it is likely that these environmental parameters have influenced the group A composition. Additionally, the stations included in group B were located around the continental slope region. The taxa that showed relatively high abundance in group B are known to prefer relatively warm water temperatures and are dominant in the open sea [36,37]. Group C also showed a relatively high abundance of open-sea taxa. Given that the northwest Ross Sea region is one of the pathways for MCDW, these oceanic taxa flowed into the Ross Sea continental shelf region around the northwest slope region in January through MCDW [38]. Additionally, Cirriped nauplius occupied almost 40% of the total abundance in group C. However, this resulted in unusually high abundance in station B41, not following the overall pattern of group C. On the other hand, group D showed higher *Euphausia* calyptopis abundance than the other groups. Considering that ice krill spawn in the continental shelf region in December, these larvae are considered ice krill larvae [3,39,40]. This high abundance results from the hatching of eggs in December and their development. Additionally, this result corresponds well with the chlorophyll-*a* concentration in the continental shelf. This high chlorophyll-*a* concentration favors the survival of larval krill in the Ross Sea continental shelf region in January [40–42]. Considering the divided groups with the environmental parameters, although the mesozooplankton community was similar to group D, the number of open-sea taxa was higher in group C than in group D. Overall, open-sea taxa advected into the Ross Sea continental shelf region as MCDW flowed into the Ross Sea continental shelf region in January. However, larval ice krill occupied the dominant position in the southern part of the Ross Sea continental shelf, spawning in December. Consequently, the mesozooplankton community structure in the Ross Sea continental shelf region in January was influenced by the spawning of ice krill and the flow of MCDW into the continental shelf region. According to [41], seasonal dynamics affect the summer mesozooplankton community in the Ross Sea. Furthermore, these distribution patterns could change with global warming, as water temperatures would increase, and the open-sea taxa would expand their distribution to the south [20,42]. Therefore, further research on the distribution patterns in the summer season is required to adequately monitor the Ross Sea region Marine Protected Area (RSR MPA).

**Supplementary Materials:** The following supporting information can be downloaded at https://www.mdpi.com/article/10.3390/d16030174/s1: Table S1: Species list and abundance (individuals/m$^3$) of the mesozooplankton of the eleven stations (B004 to B44) in the Ross Sea region Marine Protected Area (RSR MPA); Table S2: Species list and abundance (individuals/m$^3$) of the mesozooplankton of the eleven stations (B046 to B80) in the Ross Sea region Marine Protected Area (RSR MPA); Table S3: SIMPER analysis results showing species that contribute to group similarity; Table S4: SIMPER analysis results representing species that contribute to the dissimilarity between each group; Figure S1: Sea ice concentration from 19 January 2023 to 29 January 2023 derived from AMSR-E/AMSR2.

**Author Contributions:** Conceptualization, J.-H.K. and H.S.L.; methodology, S.H.K. and W.S.; software, S.H.K.; formal analysis, S.H.K.; data curation, S.H.K. and H.S.L.; writing—original draft preparation, S.H.K.; writing—review and editing, S.H.K., W.S., J.-H.K. and H.S.L.; visualization, S.H.K.; supervision, H.S.L.; project administration, J.-H.K.; funding acquisition, J.-H.K. All authors have read and agreed to the published version of the manuscript.

**Funding:** This research was funded by a Korea Institute of Marine Science & Technology Promotion (KIMST) grant funded by the Ministry of Oceans and Fisheries (KIMST 20220547) and partially supported by a grant from the Korea Polar Research Institute (PE24110).

**Institutional Review Board Statement:** Not applicable.

**Data Availability Statement:** All data are available at the Korea Polar Data Center. Requests for further data can be directed to the corresponding author.

**Acknowledgments:** The authors thank all those involved in sample collection for their contributions. The authors appreciate the assistance of the captain and crew of IBRV Araon.

**Conflicts of Interest:** The authors declare no conflicts of interest.

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
