# Peer review of "Spatial Distribution Pattern of the Mesozooplankton Community in Ross Sea Region Marine Protected Area (RSR MPA) during Summer"

_diversity, doi:10.3390/d16030174_

Round 1
Reviewer 1 Report
Comments and Suggestions for Authors
Manuscript ID: diversity-2887539
Type of manuscript: Article
Title: Spatial distribution pattern of the mesozooplankton community in
January Ross Sea region Marine Protected Area (RSR MPA)
Authors: Sung Hoon Kim, Wuju Son, Jeong-Hoon Kim, And Hyoung Sul La
The work is devoted to one of the most remote regions of the World Ocean. The very fact that this study has been realized in that region makes it valuable and meriting publication in Diversity journal. Analysis of the large-scale zooplankton distribution and its correlation with environmental parameters adds value to this investigation. The statistical methods used are adequate to the goal and tasks of the study. However, poor language prevents the reader from perceiving the results of the study. I strongly recommend to address the native speaker or language service to improve this side of the work!
Further comments and the minor corrections are presented below.
Abstract
L. 14 – "of the January RSR MPA" – better "in the RSR MPA in January"
L. 17 – "In hierarchical analysis" – better "As hierarchical analysis has shown"
L. 19 – "composed" = "was composed"
- Are these groups only variants of the same community, or different communities, with different set of dominants?
Introduction
L. 34 – "gather around" – better "attract (higher trophic level organisms)"
The Introduction is too short. It would be nice if authors analysed shortly the data received before on the zooplankton of the Ross Sea (qualitative and quantitative differences between areas studied), not only presented references to the previous studies.
Environmental factors are main drivers of organism distribution – here they are mentioned as something additional, not so important. I would recommend authors to highlight the priority of factors as drivers of plankton distribution. Geography is just "canvas" for a picture drawn by factors. Only temperature is mentioned of factors studied, while salinity and nutrients were measured as well.
Materials and Methods
L. 58 – mesozooplankton was collected, but environmental factors were measured; "As a result" – unnecessary words.
L. 70 – Why potential temperature was used in analysis? It is unusual for biological works, so it is difficult to compare with other studies. If there is some reason, explain please.
L. 80 – "Trilogy" – it is not a type of instrument, just model name. It is fluorometer.
L. 84 – "Auti Analyzer" – maybe "autoanalyser"?
L. 88 – Why cholorophyll a data was averaged over 100 m, while other – over 200 m? Please explain.
L. 102-104 – The sentence is difficult to apprehend.
- It is not clear whether sampling was conducted in polynyas or from under the ice, or both.
Results
L. 110 – "environmental values" – " environmental parameters". And further in the text.
L. 111 – "environmental values of the sampling stations" – "The values of environmental parameters at the sampling stations".
Table 1. – There are not measurement units indicated. They should be described also in methodological section.
Fig. 2 – The size of font, or the figure as a whole should be increased. And units are not indicated.
Fig. 3 – What units of the chlorophyll quantity are used (1-st panel)?
L. 152 – "Cyclopods" – Cyclopoida.
L. 161-163 – The sentence is very difficult to perceive. Please, check syntax!
Fig. 4 – Please, enlarge the font in legends.
Fig. 5 – Enlarge the font in the panel B.
L. 192-210 – The whole paragraph is very difficult to perceive. All the information, mentioned in the paragraph is present in tables. I recommend to confine this part of text to general conclusions from the table.
L. 221 – Misprint: Figure 6A is meant here. And it looks like most taxa correlate positively with group B, not A and C; Euphausia calyptopis and Limacina rangii correlate with group D; Fish larvae correlate with groups A and C.
L. 224 - Misprint: Figure 6B is meant here.
Fig. 6 – The font in figure is too small. Here the size of the whole figure could be increased. Dots of stations would be helpful in the panel B.
Discussion
CDW and MCDW – abbreviations are not deciphered before.
L. 256 – The sentence is unclear. And further the language is very difficult to perceive.
L. 264 - "cyclopods" – Cyclopoida
L. 265 – Low abundances of Oithona and Triconia are for sure the consequences of the large mesh use! The 200 µm mesh in our study undersamples about 90% of these species populations!
L. 268 – "four groups" – maybe "fourth group"?
L. 295 – "occupied the dominant composition" – "occupied dominant position"?
Comments on the Quality of English LanguagePoor language prevents the reader from perceiving the results of the study. I strongly recommend to address the native speaker or language service to improve this side of the work!
Reviewer 2 Report
Comments and Suggestions for Authors
Review Kim et al. ‘Spatial distribution pattern of the mesozooplankton community in January Ross Sea region Marine Protected Area (RSR MPA).
The manuscript analyses the composition of the zooplankton in the Antarctic Ross Sea in relation to environmental factors. Data on the zooplankton composition in such remote areas is generally rare and the paper provides new and valuable about changes of diversity across the shelf. In this respect, I advise that the paper should be published in ‘Diversity’. There are, however, several issues that need to be resolved.
The style and language needs to be considerably improved. There re incomplete sentences in which verbs are missing, typos or the use of incorrect wording throughout the manuscript. See, for instance line 30-35, which illustrates the problem best. ‘However’ is used not always in the correct context. I recommend checking by a native speaker or editing service.
Acronyms need explanations the first time they are used.
The title may specify the season rather than a specific month of the investigation.
The introduction should include some more scientific background of the state of the art of the knowledge about biodiversity in the area and provide some scientific objectives.
The stations numbers and legend in Figure 1 are by far too small and can barely be read. This applies also to Fig 2,3.
The methods lack some detail. How was the filtered volume of the net calculated? Add 'µm' following 505. Both nets? Why were the samples split? Were both fractions be analysed and/or were these fractions always analysed as a whole? Literature for identification? Provide the depth of sampling with Niskin bottles and correct the order of filtrations for Chl a. The lack of filtration of samples for nutrient analysis before storage for several days likely bias measurements. For the interpretation of the zooplankton distribution, nutrient data is not strictly necessary.
Data processing: How was normalizing achieved?
Table 3-4 should be moved to the supplement.
Comments on the Quality of English Languagesee comments to the authors; theer are too many corrections necessary to be listed here
Reviewer 3 Report
Comments and Suggestions for Authors
The authors (hereafter, "you") of this manuscript describes a summertime study conducted across multiple stations in the remote Ross Sea Region MPA. By the first look, the content of the manuscript may not appeal "novel" to a reader/reviewer/editor, but it is rather a descriptive study of the region. Such descriptive research efforts must be deeply appreciated by the scientific community primarily because of the logistic challenges of sampling these remote waters irrespective of the season of the year. Nonetheless, I would like to see a number of improvements in the manuscript content before suggesting it for eventual publication. These comments are in the attached .PDF file. I wish you all the best with the revision.

The language needs some improvement. I have marked several instances where tuning/changing is needed in the writing. Nonetheless, there may be other instances where you need to rephrase/re-word in order to make sentences clear and/or grammatically accurate. Please consult an English language proof reader and get the manuscript tuned before submitting the revised version.
Reviewer 4 Report
Comments and Suggestions for Authors
In this study, the Authors investigated the mesozooplankton community structure and spatial distribution in Ross Sea region Marine Protected Area during January of 2023. Zooplankton samples were collected from 22 stations with bongo net together with water samples for chlorophyl and nutrients concentration analyses and measurements of environmental parameters.
The authors identified 37 mesozooplankton taxa, with seven taxa occupying almost 84% of total abundance, and Metridia gerlachei being the most abundant. Taking it all together, copepods were the most abundant group, being responsible for almost 70% of total abundance. The authors also divided mesozooplankton community into four groups based on hierarchical analysis, each associated with a specific geographical distribution. These groupings were related to various environmental factors.
The manuscript well written, and correctly structured, methods are sufficiently explained, and results are also clearly presented. Manuscript provides insight into mesozooplankton distribution pattern in the January Ross Sea continental shelf region, its relation to environmental factors and, fulfil the monitoring requirements for this marine Protected Area.
Specific remarks:
Abstract:
Line 14 “um” should it be “µm”?
Keywords:
I would suggest avoiding keywords that overlap with the title, the appropriate selection of keywords can significantly increase the visibility of the article on the Internet.
Introduction
I find this chapter to be somewhat lacking. I would appreciate authors to include more detailed description of the Ross Sea, especially for people not familiar with that region. Additional information about surface currents could also prove useful, especially in context of mesozooplankton groupings showed by the authors.
The Authors should also add a clear description of the purpose of the study.
Line 34-35 I think there is something missing in this sentence, or it should be rephrased.
Line 35 Missing space
Materials and Methods
I would ask the authors to provide more details about taxonomic identification. What stages of development have been identified and gender determined? What taxa have been identified? It's a shame that the ostracods and larvae weren't identified to the species level, as this could have been an interesting addition.
Line 64 missing unit of the mesh size.
Line 72 "CTD probe" would probably be better.
Results
Please check text alignment.
Line 113 Should this be station B18?
Line 116 Again, station B71, PSU is not a unit.
Line 147 Supplementary materials seem to be missing.
Line 157 Cirriped nauplius – I think it should be Cirripede nauplii instead.
Line 173 I’m quite sure that it’s percent of similarity, so % should be added.
Line 195 Unnecessary space in 50.05%, and before % in 50%. Please check rest of the manuscript.
I would be nice if the Authors include more tests regarding environmental values., as it is all those data seem to be underused. For example, specify which factor contribute the most in mesozooplankton distribution pattern?
Discussion
Line 270 Is it supported by taxonomic identification? I fail to see krill species being mentioned earlier in the manuscript.
Line 298-301 It may be beneficial if authors ad some information about climate change observed in this region earlier in the Introduction, as it would readers some background for this claim.
The referee
Comments on the Quality of English LanguageThe English is mostly correct, but some parts need corrections and editing. I suggest that the manuscript should be checked by a native speaker.
Round 2
Reviewer 1 Report
Comments and Suggestions for Authors
I have made some edits in the text of author's responses, which highlighted by red color (author's variant in brackets and/or highlighted by the blue color).

I strongly recommend thorough language editing.
Author Response
Comments 1: Revise the sentence.
ïƒ Response comments 1: As reviewer’s comment, the sentence was changed (Line 70).
“Furthermore, the mesozooplankton community is assessed alongside environmental factors such as temperature, salinity, and nutrients to reveal reveal the relationships between them.”
Comments 2: Revise the sentence.
ïƒ Response comments 3: As reviewer’s suggestion, the sentence was revised (Line 142).
“Considering sea ice concentration, Terra Nova Bay was covered by sea ice, while the Ross Sea was not (Figure 1S).”
”
Comments 3: Revised the title of the Table 1.
ïƒ Response comments 3: As reviewer’s suggestion, the sentence was revised (Line 170).
“Table 1. Summary of the environmental parameters sampled during the survey in the RSR MPA. Min values mean lowest values among all sampling stations, while Max values mean highest values.”
Comments 4: Revise the sentence.
ïƒ Response comments 4: As reviewer’s comment, the sentence was changed (Line 227).
“The dissimilarity between the divided groups ranged from 49.23% to 69.26% (Table S4).”
Comments 5: Revise the sentence.
ïƒ Response comments 5: As reviewer’s suggestion, the sentence was changed (Lines 281‒282).
“Among 37 mesozooplankton taxa, 7 had abundance exceeding 1 ind./m3 and constituted about 84% of the total zooplankton abundance.”
Comments 6: Revise the sentence.
ïƒ Response comments 6: As reviewer’s comment, the sentence was revised (Line 297).
“Considering the mesozooplankton community structure of the studied groups, group A, comprising the Terra Nova Bay stations, showed relatively low abundance except for station B16.”
Comments 7: I strongly recommend thorough language editing.
ïƒ Response comments 7: As you suggested the language editing has been carried out by MDPI editing service during the first revision.
Reviewer 3 Report
Comments and Suggestions for Authors
I thank the authors for addressing all the comments in a short time.
Author Response
We did our best to revise the manuscript following with reviewer's comments. We do appreciate the valuable and detailed review of our manuscript.